# Theory of Allosteric Regulation in Hsp70 Molecular Chaperones

Wayne A. Hendrickson[1,2] 

[1]Department of Biochemistry and Molecular Biophysics, Columbia University, New York, NY 10032, USA and
[2]Department of Physiology and Cellular Biophysics, Columbia University, New York, NY 10032, USA

## Research Article

**Key words:**
Allostery; ATP; polypeptide client; protein folding.

**Author for correspondence:**
*Correspondence to: Wayne A. Hendrickson,
E-mail: wah2@cumc.columbia.edu

## Abstract

Heat-shock proteins of 70 kDa (Hsp70s) are ubiquitous molecular chaperones that function in protein folding as well as other vital cellular processes. They bind and hydrolyze ATP in a nucleotide-binding domain (NBD) to control the binding and release of client polypeptides in a substrate-binding domain (SBD). However, the molecular mechanism for this allosteric action has remained unclear. Here, we develop and experimentally quantify a theoretical model for Hsp70 allostery based on equilibria among Hsp70 conformational states. We postulate that, when bound to ATP, Hsp70 is in equilibrium between a restraining state (R) that restricts ATP hydrolysis and binds peptides poorly, if at all, and a stimulating state (S) that hydrolyzes ATP relatively rapidly and has high intrinsic substrate affinity but rapid binding kinetics; after the hydrolysis to ADP, NBD and SBD disengage into an uncoupled state (U) that binds peptide substrates tightly, but now with slow kinetics of exchange.

## Introduction

Hsp70 proteins are preeminent among molecular chaperones in that their actions also feed into Hsp60, Hsp90 and Hsp100 chaperone systems and into protein degradation systems (Hartl *et al.*, 2011). They participate in diverse cellular processes; going beyond namesake stress responses, they play crucial roles in normal cells for protein folding, disassembly, degradation and membrane translocation, and they are protective against neurodegenerative diseases (Ciechanover and Kwon, 2017) and complicit in cancers (Murphy, 2013). Hsp70s are found in all forms of life, excepting certain archaea, and in all ATP-containing cellular compartments of eukaryotes. Their sequences are highly conserved (>40% pairwise amino-acid identity), and especially so within three subfamilies corresponding to the eukaryotic cytosol, to the endoplasmic reticulum (ER), and to mitochondria, chloroplasts and prokaryotes. Although many Hsp70s are stress-induced, others are expressed constitutively from essential genes (Daugaard *et al.*, 2007).

Hsp70 proteins act in ATP-dependent cycles of binding and release of client substrates, typically exposed hydrophobic polypeptide segments. ATP binding and hydrolysis occurs in the nucleotide-binding domain (NBD), which controls the binding and release of polypeptides in the substrate-binding domain (SBD) (Zuiderweg *et al.*, 2013; Mayer and Kityk, 2015). The binding functions of NBD and SBD are separable, but Hsp70 chaperone activity requires direct, albeit transient, allosteric interactions between these sites as linked together (Hartl *et al.*, 2011; Zuiderweg *et al.*, 2013; Mayer and Kityk, 2015). ATP binding to NBD dramatically decreases SBD affinity for client substrates. Reciprocally, substrate binding stimulates ATP hydrolysis, whereupon substrates are retained tightly bound. Both on and off rates for substrate binding are accelerated in the presence of ATP relative to that with ADP-bound or nucleotide-free states. Hsp40s further stimulate ATP hydrolysis by Hsp70s and help to target them to substrates, and Hsp110s and other nucleotide exchange factors (NEFs) facilitate the release of ADP and rebinding of ATP; nevertheless, the Hsp70 chaperone cycle can proceed *in vitro* without these cofactors. The current picture of Hsp70 function is consistent with initial suggestions of Pelham (1986) and Rothman (1989) that molecular chaperones bind to aggregation-prone surfaces induced by stress and employ the energy of ATP hydrolysis for staged release and folding.

Crystal structures of individual NBD and SBD domains of Hsp70s provide a framework for biochemical understanding of Hsp70 chaperone activity. The prototype NBD structure is that from bovine Hsc70 (bHsc70) (Flaherty *et al.*, 1990). It comprises four subdomains (IA, IB, IIA and IIB) built up from two structurally similar lobes (I and II). Adenosine nucleotides bind at the interface between the lobes, making contacts with all four subdomains. The prototype SBD structure is that from Hsp70 DnaK of *Escherichia coli* (Zhu *et al.*, 1996). A substrate peptide is bound in an extended conformation through a channel defined by loops from SBDβ and covered by the SBDα subdomain. Findings from these prototypical structures were extended in numerous biochemical and biophysical studies as reviewed (Hartl *et al.*, 2011; Zuiderweg *et al.*, 2013; Mayer and Kityk, 2015).

Interactions between NBD and SBD are clearly essential for allosteric communication between the nucleotide and peptide-binding sites in an Hsp70 chaperone; however, the contacts are labile and their capture has proved elusive. For example, early efforts to crystallize full-length Hsp70s with ATP led instead to the structure of an NBD–ADP complex after ATP hydrolysis and incidental proteolysis (Sriram et al., 1997). NBD and SBD are flexibly linked when in ADP or nucleotide-free states (Bertelsen et al., 2009), and crystal structures that are obtained often have the DLLLLD-like NBD–SBD linker segment engaged adventitiously with a lattice mate (Chang et al., 2008; Adell et al., 2018). ATP hydrolysis is typically too facile for the capture of stable Hsp70-ATP complexes, although NBD–linker–SBD interactions are evident when ATP is bound to an Hsp70 (Zhuravleva and Gierasch, 2011), or even simply to an NBD–linker construct (Vogel et al., 2006; Swain et al., 2007). Fortunately, yeast Hsp110 Sse1 provided us with a plausible Hsp70-ATP model (Liu and Hendrickson, 2007). Hsp110s bind but do not hydrolyze ATP, yet their sequences include unmistakable NBD domains and remote resemblances in SBD domains. The structure of Sse1-ATP showed interfaces between redisposed domains that are extensive, intimately complementary, and engaging of residues strikingly conserved in classic Hsp70 sequences. These features suggested that the Sse1-ATP interfaces might be evolutionary vestiges of functional Hsp70 interfaces. In confirmation, in vivo mutational tests of the inferred interfaces in yeast Hsp70 Ssa1 and E. coli Hsp70 DnaK produced severe phenotypes in each at 8 of 9 tested interfacial contact sites including 10 of 13 specific mutations (Liu and Hendrickson, 2007).

The Hsp110 structure inspired the design of DnaK constructs that could mimic the conformation found in the Sse1 structure and, using the hydrolysis-impaired T199A mutant (Barthel et al., 2001), confirmatory structures were obtained for ATP complexes (Kityk et al., 2012; Qi et al., 2013). Subsequently, this conformational state of Hsp70-ATP was corroborated in structures of human Hsp70-8 (BiP) (Yang et al., 2015) and yeast SsaB (Gumiero et al., 2016). Further biochemical tests on selected interface DnaK mutants (Wang et al., 2020) led us to speculate that this Hsp110-like conformation must be restrained against ATP hydrolysis and essentially devoid of ATP binding, which in turn begged the questions of how the rebinding of substrate peptides and ATP hydrolysis might occur and how restraints against hydrolysis are effected. Additional structural analyses have shed light on these questions (Wang et al., 2020; Wang and Hendrickson, 2020a; Wang and Hendrickson, 2020b).

There is a rich literature from previous theoretical investigations of allosteric interactions in proteins (Monod et al., 1965; Koshland et al., 1966; Cui and Karplus, 2008; Motlagh et al., 2014; Cuendet et al., 2016; Thirumalai et al., 2019). Notably, the influential allosteric theories of Monod, Wyman and Changeux (MWC) (Monod et al., 1965) and of Koshland, Némethy and Filmer (KMF) (Koshland et al., 1966) have treated cooperativity between similar binding sites in symmetric oligomers such as hemoglobin. The particular treatments of MWC and KMF are not directly applicable to Hsp70s, however; as these are monomeric proteins, predominantly, with distinct binding domains for altogether different ligands. What does apply more broadly is the concept that conformational equilibria between alternative states can govern allosteric regulation, and this has been pursued productively (Cui and Karplus, 2008; Motlagh et al., 2014; Cuendet et al., 2016; Thirumalai et al., 2019). Nevertheless, we are not aware of quantitative treatments in the MWC mode for allostery in Hsp70 systems.

In an attempt to understand the distinctive Hsp70 mechanisms for allosteric control, we have developed a theoretical model for equilibria among conformational states in Hsp70 chaperones. This theory explains observations on ATP hydrolysis and polypeptide binding from wild-type (WT) and mutant variant DnaKs by postulating that ATP-bound Hsp70s equilibrate between states with distinct characteristics for the binding of substrate peptides and for the hydrolysis of ATP. Our biochemical results are fitted quantitatively by this allosteric theory, and the postulated but previously uncharacterized stimulating state has now been confirmed by crystal structures (Wang et al., 2020).

## Theory

We assume that an Hsp70 chaperone protein exists in an equilibrium of states. Its NBD may bind ATP, ADP or be nucleotide-free (which we denote as Apo), and its SBD may bind segments of substrate polypeptides. The population of Hsp70 molecules that are complexed with ATP are in equilibrium between a restraining state, which binds substrate peptides poorly at best and only hydrolyzes ATP at a low basal rate, and a stimulating state, which binds substrate peptide well and hydrolyzes ATP at a substantially elevated rate. After ATP hydrolysis to the ADP state, peptide substrates are retained with high affinity in a state without allosteric coupling between the nucleotide and peptide binding sites. ADP may dissociate to yield the Apo state, remaining allosterically uncoupled and retaining the substrate peptide if present. Hsp70-Apo may rebind ATP to reinitiate the chaperone cycle. Additional intermediate states may also exist.

### Overview of equilibrating states and analytic approach

The allosteric interactions between ATP in the NBD domain and a client peptide in the SBD domain can be followed biochemically by measuring peptide binding in the presence ATP and by measuring ATP hydrolysis in the presence of a client peptide. Such biochemical measurements on Sse1-inspired mutants provoked us to contemplate the theoretical basis for Hsp70 allostery (Wang et al., 2020), attempting to explain the observations as consequences of an equilibrium between restraining and stimulating states. We identify the restraining state as $Hsp70_R$-ATP, abbreviated as R, and the stimulating state as $Hsp70_S$-ATP, S for short. Both of these are ATP-bound states having NBD and SBD engaged for inter-site communication, as observed for R(Kityk et al., 2012; Qi et al., 2013 ) and proposed for S, whereas the binding domains are flexibly linked and thereby uncoupled in ADP and Apo states (Bertelsen et al., 2009).

We first studied a model featuring an R state that cannot bind client peptides and hydrolyzes ATP at a basal rate being in equilibrium with an S state that can bind peptides to form the SP state, with both S and SP hydrolyzing ATP at a more elevated rate. We found that hydrolysis data were fitted well by this model, but that resulting parameters underestimated the apparent peptide affinity in ATP. We then tested a model that allowed for client peptide binding to R as well as S, but found such binding to be incompatible with the hydrolysis data. Finally, we elaborated an alternative model to include a quasi-intermediate conformation Q that can bind peptides as in S but hydrolyzes ATP at the basal rate as in R, but now with an R that cannot bind peptides.

The equilibria and hydrolytic reactions relating the various states in these alternative models are illustrated schematically in

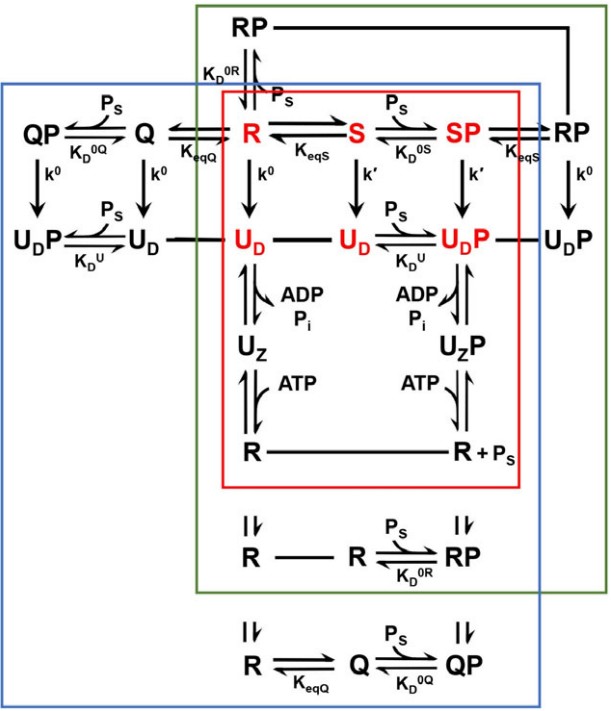

**Fig. 1.** Network of Hsp70 equilibria and hydrolytic reactions. Symbols S, R, Q, $U_D$ and $U_Z$ correspond to the stimulating, restraining, quasi-intermediate, uncoupled ADP and uncoupled Apo states, respectively, and SP, RP, QP $U_DP$ and $U_ZP$ are the corresponding peptide complexes. States are shown connected by equilibria, reactions or single-line designations of identity (——). Equilibria constants $K_{eqS}$, $K_D^{0S}$, $K_D^{0R}$, $K_D^U$, $K_D^{ADP}$, $K_D^{ATP}$, $K_{eqQ}$ and $K_D^{0Q}$, are defined by Eqs. (1)–(6) and (34), respectively, and catalytic rate constants $k'$ and $k^0$ are defined by Eqs. (13) and (14), respectively. Core exchanges of the allosteric system involve only R, S, SP, U and UP states, and these are indicated by red symbols and are contained in the red box, where substrate peptide $P_S$ is expelled upon ATP binding to $U_ZP$. Exchanges that also include RP are contained in the green box, where ATP binding to $U_ZP$ yields RP. Exchanges that also include Q and QP are contained in the blue box, where ATP binding to $U_ZP$ yields QP. The red core is the subset of the blue set with no peptide binding to R ($K_D^{0R} = \infty$).

Fig. 1, and mathematical details are described in following sections. The system is complex, even as here stripped of cofactors and partner chaperones. These models are neither fully comprehensive nor singularly unique, and the number of parameters may challenge experimental evaluation. Nevertheless, the theory is providing useful insights as found from experimental validations described below and as used in designing constructs that captured the postulated S state (Wang et al., 2020).

Our analytic approach is to evaluate the rate of ATP hydrolysis ($k_{cat}$) as measured in single-turnover reactions at steady state (Wang et al., 2020; Davis et al., 1999) and the apparent dissociation constant for peptide binding ($K_D^{APP}$) in the presence of ATP, both as functions of hydrolysis rates and intrinsic equilibrium constants (Fig. 1). Because of the hydrolytic reactions, the system cannot be analyzed usefully at equilibrium; however, the steady-state analysis is feasible since Hsp70-catalyzed ATP hydrolysis is relatively slow ($k^0 = 0.0075$ min$^{-1}$ and $k' = 0.276$ min$^{-1}$; DnaK at 20°C, Wang et al., 2020) compared to the kinetics of relevant conformational changes. This allows the distribution of species to equilibrate at any instant. We do not know the dynamics of R-to-S exchange; however, measurements have been reported for kinetics of the uncoupled (U)-to-R conformational changes (498 min$^{-1}$ for SBDα, 1,656 min$^{-1}$ for SBDβ, and 6,420 min$^{-1}$ for the NBD–SBD linker;

DnaK at 30°C; Kityk et al., 2012) and for peptide binding and release ($k_{on} = 198$ min$^{-1}$ and $k_{off} = 1.44 \times 10^6$ min$^{-1}$ M$^{-1}$, i.e. 2.9 min$^{-1}$ at 2 μM DnaK; DnaK plus Cro peptide at 25°C; Slepenkov and Witt, 1998).

### Allosteric coupling of substrate-peptide binding to ATP hydrolysis

The equilibria between ATP states are described by Eqs. (1)–(3), where $K_{eqS}$ is the equilibrium constant between R and S states, $K_D^{0S}$ is the dissociation constant that relates the complex SP of Hsp70$_S$-ATP with a substrate peptide P to its dissociated constituents, S and P, and $K_D^{0R}$ is the dissociation constant that relates the complex RP between Hsp70$_R$-ATP and a substrate peptide P to its dissociated products, R and P. The corresponding kinetic rates for peptide association and dissociation are specified as $k_{on}^S$, $k_{off}^S$, $k_{on}^R$ and $k_{off}^R$.

$$K_{eqS} = \frac{[R]}{[S]}, \tag{1}$$

$$K_D^{0S} = \frac{[S][P]}{[SP]} = \frac{k_{off}^S}{k_{on}^S}, \tag{2}$$

$$K_D^{0R} = \frac{[R][P]}{[RP]} = \frac{k_{off}^R}{k_{on}^R}. \tag{3}$$

### Substrate-peptide binding to allosterically uncoupled ADP and Apo Hsp70s

We postulate that Hsp70$_S$-ATP hydrolyzes ATP to Hsp70$_U$-ADP (U) and SP hydrolyzes ATP to yield UP, both at the rate of $k'$ whether peptide is bound or not, and Hsp70$_R$-ATP states, R and RP, both hydrolyze ATP at the basal rate of $k^0$ to produce the respective ADP states U and UP. U and UP are related by the dissociation equilibrium Eq. (4), which is defined by a dissociation constant $K_D^U$ that comprises the kinetic constants $k_{on}^U$ and $k_{off}^U$. We expect the constants of Eq. (4) to be the same for any state of SBD that is uncoupled (hence the superscript U) from NBD, including isolated SBD or nucleotide-free Hsp70 as well as Hsp70$_U$-ADP:

$$K_D^U = \frac{[U][P]}{[UP]} = \frac{k_{off}^U}{k_{on}^U}. \tag{4}$$

This reaction does not affect directly the reactions associated with the ATP-bound states, and it is not needed further for this analysis of allosteric interactions.

### ADP dissociation from Hsp70 and ATP rebinding to this Apo state

According with observation, we postulate that ADP can dissociate from Hsp70$_U$-ADP (D) to yield Hsp70$_U$-Apo as related by Eq. (5):

$$K_D^{ADP} = \frac{[Hsp70_U - Apo][ADP]}{[Hsp70_U - ADP]} \tag{5}$$

and that re-association of ATP with Hsp70$_U$-Apo is governed by Eq. (6):

$$K_D^{ATP} = \frac{[Hsp70_U - Apo][ATP]}{[Hsp70_U - ATP]}. \tag{6}$$

The inorganic phosphate ($P_i$) product of ATP hydrolysis remains Hsp70-bound with ADP after ATP hydrolysis (Flaherty *et al.*, 1990; Sriram *et al.*, 1997; Wang and Hendrickson, 2020a); thus, Eq. (5) connotes both ADP and $P_i$ release. Since NBD and SBD are presumed uncoupled in the ADP and Apo states, Eqs. (5) and (6) apply equally to peptide-bound and peptide-free states of $Hsp70_U$-ADP and $Hsp70_U$-Apo. These associations apply when Hsp70 alone can interact freely with the reactants, which typically exist with ATP in excess of ADP; however, the consequent nucleotide exchange reactions can be accelerated substantially by nucleotide exchange factors, notably GrpE for DnaK or Hsp110 for eukaryotic Hsp70s.

### Distribution of Hsp70-ATP between restraining and stimulating states

The total concentration of Hsp70-ATP, $c_T$, is given by Eq. (7):

$$c_T = [R] + [RP] + [S] + [SP]. \tag{7}$$

$c_T(t)$ changes as time proceeds because of ATP hydrolysis and possible ATP binding to the apo state or exchange of ATP for ADP; however, we assume that the exchanges governed by equilibria (1)–(3) are sufficiently rapid that $c_T(t)$ at any instant is governed by Eq. (8):

$$c_T = K_D^{0S}\left\{\left(K_{eqS}\left(1 + [P]/K_D^{0R}\right) + 1 + [P]/K_D^{0S}\right)\right\}[SP]/[P] \tag{8}$$

or

$$[SP] = \frac{c_T[P]}{K_D^{0S}\left(K_{eqS} + 1\right) + [P]\left\{1 + \left(K_D^{0S}/K_D^{0R}\right)K_{eqS}\right\}}. \tag{9}$$

From Eq. (7), the fraction $Q_S$ of the total Hsp70-ATP protein that is in the stimulating state is given by Eq. (10):

$$Q_S = \frac{[S] + [SP]}{[R] + [RP] + [S] + [SP]} \tag{10}$$

and in light of Eqs. (1) and (2), this yields

$$Q_S([P]) = \frac{[P] + K_D^{0S}}{[P]\left(1 + \left(K_D^{0S}/K_D^{0R}\right)K_{eqS}\right) + K_D^{0S}\left(K_{eqS} + 1\right)}. \tag{11}$$

And for the case of $K_D^{0R} = \infty$, the limit of no peptide binding to the restraining state,

$$Q_S([P]) = \frac{[P] + K_D^{0S}}{[P] + K_D^{0S}\left(K_{eqS} + 1\right)}. \tag{12}$$

### ATP hydrolysis in the presence of a peptide substrate

We first wish to understand the allosteric control of ATP hydrolysis by peptide binding. We postulate that $Hsp70_S$-ATP hydrolyzes ATP to ADP and $P_i$ at the same rate, $k'$, whether complexed with peptide or not, in state SP or S. We similarly postulate that $Hsp70_R$-ATP hydrolyzes ATP to ADP and $P_i$ at its own rate, $k^0$, again whether complexed with peptide or not, that is as RP or R. These reactions are designated in Eqs. (13) and (14):

$$S\,(ATP) \xrightarrow{k'} U\,(ADP), \qquad SP\,(ATP) \xrightarrow{k'} UP\,(ADP), \tag{13}$$

$$R\,(ATP) \xrightarrow{k^0} U\,(ADP), \qquad RP\,(ATP) \xrightarrow{k^0} UP\,(ADP). \tag{14}$$

For ATP hydrolysis, each state contributes to the observed catalytic rate in proportion to its relative abundance and the associated rate of hydrolysis as described by Eq. (15):

$$k_{cat} = \left\{ [R]k^0 + [RP]k^0 + [S]k' + [SP]k' \right\}/c_T. \tag{15}$$

Taking Eq. (7) into account, Eq. (15) yields

$$k_{cat} = \left\{ (c_T - [S] - [SP])k^0 + ([S] + [SP])k' \right\}/c_T$$

$$k_{cat} = k^0 + ([S] + [SP])\left(k' - k^0\right)/c_T.$$

With reference to Eq. (2), we obtain Eq. (16):

$$k_{cat} = k^0 + \left([P] + K_D^{0S}\right)\left(k' - k^0\right)[SP]/(c_T[P]). \tag{16}$$

Then, upon substitution of Eq. (9) into Eq. (16),

$$k_{cat} = k^0 + \frac{[P] + K_D^{0S}}{[P]\left(1 + \left(K_D^{0S}/K_D^{0R}\right)K_{eqS}\right) + K_D^{0S}\left(K_{eqS} + 1\right)} \times \left(k' - k^0\right). \tag{17}$$

Equation 17 can be rearranged to yield the mathematically explicit form

$$k_{cat} = \frac{[P]\left(k' + \left(K_D^{0S}/K_D^{0R}\right)K_{eqS}k^0\right) + K_D^{0S}\left(K_{eqS}k^0 + k'\right)}{[P]\left(1 + \left(K_D^{0S}/K_D^{0R}\right)K_{eqS}\right) + K_D^{0S}\left(K_{eqS} + 1\right)}$$

$$= \frac{a\,[P] + b}{c\,[P] + d}. \tag{18}$$

A particular case, which was in fact motivating to our analysis, arises when the restraining state has no affinity for peptide substrates; that is when $K_D^{0R} = \infty$. Then, Eqs. (17) and (18) reduce to

$$k_{cat} = k^0 + \frac{[P] + K_D^{0S}}{[P] + K_D^{0S}\left(K_{eqS} + 1\right)}\;x\;\left(k' - k^0\right), \tag{19}$$

and

$$k_{cat} = \frac{k'[P] + K_D^{0S}\left(K_{eqS}k^0 + k'\right)}{[P] + K_D^{0S}\left(K_{eqS} + 1\right)} = \frac{a\,[P] + b}{[P] + d}. \tag{20}$$

### Degeneracy in allosteric parameters

The mathematical form of Eq. (20) has only three independent parameters: $a = k'$, $b = K_D^{0S}\left(K_{eqS}k^0 + k'\right)$, and $d = K_D^{0S}\left(K_{eqS} + 1\right)$ even though the theory is formulated in terms of four physically meaningful parameters, even as simplified by ignoring peptide binding in the restraining state as for Eqs. (17) and (18). Thus, a degeneracy in solutions must arise from the fitting of measurements of hydrolytic rate $k_{cat}$ at varied peptide concentrations $[P]$; $k'$ is determined uniquely, but only the $b$ and $d$ combinations of other physical parameters are determined uniquely. In principle, after either $K_D^{0S}$ or $K_{eqS}$ is specified, or in certain special cases (e.g. $k^0 = 0$), then the other parameters can be separated. In practice, we break the degeneracy by fixing one parameter from separate measurements. For example, in studies on Hsp70 DnaK, we are able to generate conditions that fix the protein in the stimulating state whereby peptide binding in the presence of ATP serves to define $K_D^{0S}$.

## Mathematical fitting

Parameters a, b, c and d of Eq. (18) can only be determined relative to a common factor, which is most conveniently taken as c, since c = 1 corresponds to $K_D^{0R} = \infty$ for no peptide affinity in the R state. Thereby,

$$k_{cat} = \frac{a'[P] + b'}{[P] + d'}, \qquad (21)$$

where $a' = a/c$, $b' = b/c$, $c' = 1$ and $d' = d/c$. The formalism of Eq. (21) permits unique fitting to a set of $k_{cat}$ versus [P] data; however, biochemical interpretations can then be made for any arbitrary value of c using Eq. (18) provided that the mathematical degeneracy is broken by specifying one parameter in formalism Eq. (20), which we take here to be that for $K_D^{0S}$. Then from the coefficients of Eq. (18), we can evaluate the allosteric parameters at arbitrary scalings c:

$$\text{From} \quad d' = K_D^{0S}\left(K_{eqS} + 1\right), K_{eqS} = \left(d/cK_D^{0S}\right) - 1. \qquad (22)$$

$$\text{From} \quad c = 1 + \left(K_D^{0S}/K_D^{0R}\right) K_{eqS}, \quad K_D^{0R} = K_{eqS}K_D^{0S}/(c-1). \qquad (23)$$

$$K_D^{0R} = \infty \quad \text{when} \quad c = 1$$

$$K_D^{0S}k' + K_D^{0S}K_{eqS}k^0 = b' \qquad (24)$$

$$k' + K_{eqS}\left(K_D^{0S}/K_D^{0R}\right) k^0 = a' \qquad (25)$$

$$K_D^{0S}k' + K_D^{0S}K_{eqS}\left(K_D^{0S}/K_D^{0R}\right)k^0 = a'K_D^{0S} \qquad (26)$$

From (24) and (26) $k^0 = \left(b - K_D^{0S}a\right)/c\, K_D^{0S}K_{eqS}\left(1 - K_D^{0S}/K_D^{0R}\right)$. $\qquad (27)$

$$k^0 = \left(b - K_D^{0S}a\right)/K_D^{0S}K_{eqS} \quad \text{when } c = 1$$

$$\text{From (25)} \quad k' = a/c - K_{eqS}\left(K_D^{0S}/K_D^{0R}\right) k^0 \qquad (28)$$

$$k' = a \qquad \text{when } K_D^{0R} = \infty \ (c = 1).$$

To summarize, given the fitting with parameters a, b and d, and an arbitrarily chosen value of c and a specified value of $K_D^{0S}$, the other biochemical parameters are determined by Eqs. (22)–(28) for $K_{eqS}$, $K_D^{0R}$, $k^0$ and $k'$, respectively.

## Alternative allosteric models for hydrolysis

In Eqs. (13) and (14), we postulate a hydrolysis model having rates of $k'$ and $k^0$ for S and R states, respectively, whether with substrate peptide or not. We can also contemplate an alternative model wherein the S state is in a restraining conformation such that its hydrolysis rate is $k^0$ until peptide binding generates SP with hydrolytic rate $k'$. More generally, S might have an arbitrary hydrolytic rate $k^S$, not necessarily either $k'$ or $k^0$. In this case, Eq. 15 will be replaced by Eq. 29:

$$k_{cat} = \left\{ [R]k^0 + [RP]k^0 + [S]k^S + [SP]k' \right\}/c_T. \qquad (29)$$

Then proceeding as from Eqs. (15) to (17)

$$k_{cat} = k^0 + \frac{[P]\left(k' - k^0\right) + K_D^{0S}\left(k^S - k^0\right)}{[P]\left(1 + \left(K_D^{0S}/K_D^{0R}\right) K_{eqS}\right) + K_D^{0S}\left(K_{eqS} + 1\right)}, \qquad (30)$$

which can be recast, as for Eq. (18) from Eq. (17), into

$$\begin{aligned} k_{cat} &= \frac{[P]\left(k' + \left(K_D^{0S}/K_D^{0R}\right) K_{eqS}k^0\right) + K_D^{0S}\left(K_{eqS}k^0 + k^S\right)}{[P]\left(1 + \left(K_D^{0S}/K_D^{0R}\right) K_{eqS}\right) + K_D^{0S}\left(K_{eqS} + 1\right)} \\ &= \frac{a\,[P] + b_S}{c\,[P] + d}. \end{aligned} \qquad (31)$$

Equations (30) and (31) reduce respectively to Eqs. (17) and (18) for $k^S = k'$, and Eq. (31) gives $b_S = b_0 = K_D^{0S}\left(K_{eqS} + 1\right)k^0$ for $k^S = k^0$.

Structural and biochemical evidence lead us to contemplate another alternative model wherein the R-state itself does not bind substrate peptides, but which is instead in equilibrium with a quasi-intermediate state Q. Q has an S-like SBD conformation, which is peptide associative, and it has an R-like NBD-SBD interface such that its hydrolysis rate is $k^0$. Thus,

$$c_T = [R] + [Q] + [QP] + [S] + [SP]. \qquad (32)$$

and

$$\begin{aligned} k_{cat} &= \left\{ [R]k^0 + [Q]k^0 + [QP]k^0 + [S]k' + [SP]k' \right\}/c_T. \\ &= k^0 + ([S] + [SP])\left(k' - k^0\right)/c_T. \end{aligned} \qquad (33)$$

As for obtaining Eq. (16) from Eq. (15), we refer to Eqs. (1) and (2) and here also add the Q-state conformational and binding equilibria:

$$K_{eqQ} = \frac{[R]}{[Q]}; \qquad K_D^{0Q} = \frac{[Q]\,[P]}{[QP]}. \qquad (34)$$

Then,

$$\begin{aligned} c_T = K_D^{0S}&([SP]/[P]) \\ &\left\{ K_{eqS} + K_{eqS}/K_{eqQ}\left(1 + [P]/K_D^{0Q}\right) + \left(1 + [P]/K_D^{0S}\right) \right\} \end{aligned} \qquad (35)$$

and

$$k_{cat} = k^0 + \frac{[P] + K_D^{0S}}{[P]\left(1 + \left(K_D^{0S}K_{eqS}/K_D^{0Q}K_{eqQ}\right) + K_D^{0S}\left(K_{eqS} + 1\right)x\left(k' - k^0\right)\right)}. \qquad (36)$$

$$\begin{aligned} &= \frac{[P]\left(k' + \left(K_D^{0S}K_{eqS}/K_D^{0Q}K_{eqQ}\right) k^0\right) + K_D^{0S}\left(K_{eqS} + K_{eqS}/K_{eqQ}\right) k^0 + k'\right)}{[P]\left(1 + \left(K_D^{0S}K_{eqS}/K_D^{0Q}K_{eqQ}\right)\right) + K_D^{0S}\left(K_{eqS} + K_{eqS}/K_{eqQ} + 1\right)} \\ &= \frac{a\,[P] + b}{c\,[P] + d}. \end{aligned} \qquad (37)$$

## Substrate peptide binding in the presence of ATP

### Substrate model of Equations (1) to (3)

We also wish to understand the allosteric effect of ATP on substrate peptide binding. The R and S states are not differentiated in typical peptide binding experiments, whereby the apparent dissociation constant that can be observed is (38):

$$K_D^{APP}(ATP) = [P]\left([R] + [S]\right)/([RP] + [SP]), \qquad (38)$$

which by Eq. (1) gives

$$K_D^{APP}(ATP) = [P][S]\left(K_{eqS} + 1\right)/([RP] + [SP]),$$

and with reference to Eqs. (2) and (3), the apparent and intrinsic dissociation constants in ATP are then related by (39):

$$K_D^{App}(ATP) = \frac{[P][S](K_{eqS}+1)}{[P][S](K_{eqS}/K_D^{0R}+1/K_D^{0S})}$$

$$= \frac{K_D^{0R}K_D^{0S}(K_{eqS}+1)}{(K_D^{0S}K_{eqS}+K_D^{0R})} \quad (39)$$

$$K_D^{App}(ATP) = \frac{K_D^{0S}(K_{eqS}+1)}{1+K_D^{0S}K_{eqS}/K_D^{0R}}.$$

In the limit of no peptide binding to the restraining state, $K_D^{0R} = \infty$,

$$K_D^{App}(ATP) = K_D^{0S}(K_{eqS}+1). \quad (40)$$

Given a measured value for $K_D^{App}$, the corresponding value for $K_D^{0R}$ can be obtained from Eq. (39):

$$K_D^{0R} = \frac{K_D^{App}(ATP) K_D^{0S}K_{eqS}}{K_D^{0S}(K_{eqS}+1) - K_D^{App}(ATP)}. \quad (41)$$

Also from Eq. (39), when the equilibrium is entirely toward the restraining state, $K_{eqS} = \infty$, $K_D^{App}(ATP) = K_D^{0R}$; and when it is entirely toward the stimulating state, $K_{eqS} = 0$, $K_D^{App}(ATP) = K_D^{0S}$.

### Alternative peptide-binding model

As described in the analysis of ATP hydrolysis, we contemplate an alternative allosteric model in which the quasi-intermediate Q state is in equilibrium with the R and S states and in which Q binds substrate peptides but R does not. In this case, analogous to Eq. (38),

$$K_D^{App}(ATP) = [P]([R]+[Q]+[S])/([QP]+[SP]), \quad (42)$$

and then, after substitutions from Eq. (34) and from Eq. (1),

$$K_D^{App}(ATP) = [R][P](1+1/K_{eqQ}+1/K_{eqS}) \\ /[R][P](1/K_D^{0Q}K_{eqQ}+1/K_D^{0S}K_{eqS}) \quad (43)$$

or

$$K_D^{App}(ATP) = \frac{(K_{eqQ}K_{eqS}+K_{eqS}+K_{eqQ})}{K_{eqQ}K_{eqS}} \\ \times \frac{K_D^{0S}K_{eqS}K_D^{0Q}K_{eqQ}}{(K_D^{0S}K_{eqS}+K_D^{0Q}K_{eqQ})}$$

or

$$K_D^{App}(ATP) = \frac{K_D^{0S}(K_{eqS}+K_{eqS}/K_{eqQ}+1)}{(1+K_D^{0S}K_{eqS}/K_D^{0Q}K_{eqQ})}, \quad (44)$$

which reduces to (40) as $K_{eqQ} = \infty$, that is $[Q] = 0$.

As for its hydrolysis counterpart Eq. (36), the variables in Eq. (44) are too numerous for independent evaluation; however, we can fix $K_D^{0S}$ at a measured value and we can estimate $K_D^{0Q}$ in relation to that value. A third specification can come from in-parallel fitting to ATP hydrolysis data by Eq. (37). Thus, using d′ from the fitting of hydrolysis data by Eq. (21) while specifying c as the scaling factor for the evaluation of d from Eq. (37), we obtain $f = (cd'/K_D^{0S}) = K_{eqS} + K_{eqS}/K_{eqQ} + 1$; while letting $g = K_D^{App}/K_D^{0S}$ and $g' = K_D^{App}/K_D^{0Q}$, we obtain $g + g' K_{eqS}/K_{eqQ} = K_{eqS} + K_{eqS}/K_{eqQ} + 1$ from Eq. (44). These two observational equations then determine values for the relevant unknowns:

$$K_{eqS}/K_{eqQ} = (f-g)/g', \quad (45)$$

$$K_{eqS} = (f-1) - (f-g)/g'. \quad (46)$$

Since both $K_{eqS}/K_{eqQ}$ and $K_{eqS}$ must be non-negative for the solution to be physical, it follows from (45) that $d \geq K_D^{App}$ and from (46) that

$$K_D^{0Q} \leq K_D^{App}(f-1)/(f-g) = (d-K_D^{0S})/(d-K_D^{App}). \quad (47)$$

### Distribution of states in the Q-alternative model

The states R, S, Q, SP and QP in the Q-alternative model are mutually exclusive, contributing to the total Hsp70 concentration, $C_T$, as given in Eq. (32). After normalization to [SP] through equilibria defined by Eqs. (1), (2) and 34,

$$C_T = K_D^{0S}(K_{eqS} + K_{eqS}/K_{eqQ} + K_{eqS}/K_{eqQ}([P]/K_D^{0Q}) \\ + 1 + [P]/K_D^{0S})[SP]/[P]. \quad (48)$$

Then, following as from Eqs. (7) to (11) for the R/S model, we obtain the fractions $Q_S$, $Q_R$ and $Q_Q$ in the S, R and Q states, respectively, as a function of peptide concentration [P]:

$$Q_S([P]) = \\ \frac{[P]+K_D^{0S}}{[P](1+(K_D^{0S}K_{eqS}/K_D^{0Q}K_{eqQ}))+K_D^{0S}(K_{eqS}+K_{eqS}/K_{eqQ}+1)}. \quad (49)$$

$$Q_R([P]) = \\ \frac{K_D^{0S}K_{eqS}}{[P](1+(K_D^{0S}K_{eqS}/K_D^{0Q}K_{eqQ}))+K_D^{0S}(K_{eqS}+K_{eqS}/K_{eqQ}+1)}. \quad (50)$$

$$Q_Q([P]) = \\ \frac{[P](K_D^{0S}K_{eqS}/K_D^{0Q}K_{eqQ})+K_D^{0S}K_{eqS}/K_{eqQ}}{[P](1+(K_D^{0S}K_{eqS}/K_D^{0Q}K_{eqQ}))+K_D^{0S}(K_{eqS}+K_{eqS}/K_{eqQ}+1)}. \quad (51)$$

In the limit of [P] = 0,

$$Q_S(0) = 1/(K_{eqS}+K_{eqS}/K_{eqQ}+1). \quad (52)$$

$$Q_R(0) = K_{eqS}/(K_{eqS}+K_{eqS}/K_{eqQ}+1). \quad (53)$$

$$Q_Q(0) = K_{eqS}/K_{eqQ}/(K_{eqS}+K_{eqS}/K_{eqQ}+1). \quad (54)$$

### *Peptide binding profiles*

Binding characteristics ($K_D$ values) can be evaluated from profiles of the saturation of peptide binding to the protein, and calculated saturation curves can be useful for demonstration purposes. Such analyses can be performed either as a function of peptide concentration at fixed protein concentration or of protein concentration at fixed peptide concentration. The various models of conformational and binding equilibria will have different profiles, and we consider two of these here.

### Allosteric model with only S-state binding

For allosteric hydrolysis model (19), R and S are in equilibrium by $K_{eqS} = [R]/[S]$ and peptide P binds to S with intrinsic affinity governed by $K_D^{0S}$, Eq. (2), but with no affinity for R ($K_D^{0R} = \infty$).

In this case, the saturation curve y for varied peptide concentrations at a fixed total protein concentration derives from

$$y = \frac{[SP]}{[R] + [S] + [SP]} = \frac{1}{([R] + [S])/[SP] + 1}. \quad (55)$$

In light of the relevant equilibria, the total protein concentration is given by (56):

$$c_T = [R] + [S] + [SP] = [S]\left(K_{eqS} + 1 + [P]/K_D^{0S}\right)$$
$$= \left\{K_D^{0S}\left(K_{eqS} + 1\right) + [P]\right\}[SP]/[P], \quad (56)$$

whereby

$$[SP] = \frac{c_T[P]}{K_D^{0S}\left(K_{eqS} + 1\right) + [P]} \quad (57)$$

and

$$[R] + [S] = c_T - [SP] = \frac{c_T\left\{K_D^{0S}\left(K_{eqS} + 1\right) + [P]\right\} - c_T[P]}{K_D^{0S}\left(K_{eqS} + 1\right) + [P]}. \quad (58)$$

From the ratio of Eqs. (58) and (57),

$$([R] + [S])/[SP] = \frac{K_D^{0S}\left(K_{eqS} + 1\right)}{[P]}, \quad (59)$$

after substitution of Eq. (59) into Eq. (55) and rearrangement, the desired saturation curve results:

$$y = \frac{[P]}{[P] + K_D^{0S}\left(K_{eqS} + 1\right)}. \quad (60)$$

Alternatively, as in our peptide-binding experiments, one can obtain saturation curve y from varied protein concentrations at a fixed peptide concentration. The appropriate formulation for this situation derives from

$$y = \frac{[SP]}{[P] + [SP]} = \frac{1}{([P]/[SP] + 1)}. \quad (61)$$

From Eq. (56),

$$[P]/[SP] = \left\{K_D^{0S}\left(K_{eqS} + 1\right) + [P]\right\}/c_T, \quad (62)$$

and, on substitution of Eq. (62) into Eq. (61), followed by rearrangement,

$$y = \frac{c_T}{c_T + \left(K_D^{0S}\left(K_{eqS} + 1\right) + [P]\right)}. \quad (63)$$

When there is no R state, that is $K_{eqS} = 0$, saturation Eqs. (60) and (63) reduce to the respective single-component counterparts Eqs. (64) and (65):

$$y = \frac{[P]}{[P] + K_D^{0S}}, \quad (64)$$

$$y = \frac{c_T}{c_T + \left(K_D^{0S} + [P]\right)}. \quad (65)$$

### Q-alternative model

The states R, S, Q, SP and QP in the Q-alternative binding model (38) are mutually exclusive and interrelated by equilibria (1), (2) and (34). The saturation model analogous with Eq. (60) can be developed readily, but it suffices here to develop the model analogous with Eq. (63) for measurements made with fixed peptide concentration and varied total protein concentration, which derives from

$$y = \frac{[QP] + [SP]}{[P] + [QP] + [SP]}. \quad (66)$$

Proceeding as from Eqs. (61) to (65), successively evaluating ([QP] + [SP]) and [P]/[SP] and then defining $p1 = (K_D^{0S} K_{eqS})/(K_D^{0Q} K_{eqQ})$, Eq. (66) yields the Q-alternative relationship (67):

$$y = \frac{c_T}{c_T + \left\{\left(K_D^{0S}\left(K_{eqS} + K_{eqS}/K_{eqQ} + 1\right)/(1 + p1)\right\} + [P]\right.}. \quad (67)$$

### Kinetics of substrate peptide association

The kinetics of substrate peptide binding and release are complicated by having Hsp70-ATP in its two states, whether each binds peptide or not. Of course, the dissociation of prebound substrates must be contemplated even after transition to a binding-deficient state, notably SP to RP when $k_{on}^R = 0$. Here, we consider the situation where the peptide concentration [P] is much in excess of [Hsp70-ATP], such that [P] can be considered constant and absorbed into pseudo-first order $k_{on}$ rate constants. Thus,

$$S + P \xrightarrow{k_{on}^S} SP; \qquad SP \xrightarrow{k_{off}^S} S + P \quad (68)$$

$$\frac{d[S]}{dt} = \frac{d[P]}{dt} = -k_{on}^S[S] + k_{off}^S[SP]; \frac{d[SP]}{dt} = -k_{off}^S[SP]$$

and

$$R + P \xrightarrow{k_{on}^R} RP; \qquad RP \xrightarrow{k_{off}^R} R + P \quad (69)$$

$$\frac{d[R]}{dt} = \frac{d[P]}{dt} = -k_{on}^R[R] + k_{off}^R[RP]; \frac{d[RP]}{dt} = -k_{off}^R[RP].$$

The rate equations of (68) and (69) cannot be solved analytically, even for either alone. In principle, however, at equilibrium one obtains Eqs. (2) and (3), respectively, for the individual associations; and the two are linked by Eq. (1). Moreover, in the contemplated event of having no peptide binding in the R state ($k_{on}^R = 0$), [RP] = 0 at equilibrium. Nevertheless, previously bound peptide P dissociates from RP according to Eq. (69), whereupon re-association must occur via Eq. (68) after equilibration to the S state. Thus, for cases such as WT DnaK at low [P] where $K_{eqS}$ favors the R-state, $K_D^{APP}(ATP)$ may be dominated by $k_{off}^R$ and $k_{on}^S$ even when $k_{on}^R \to 0$ and $K_D^{0R} \to \infty$.

Considerations on the effect of conformational equilibria apply to reaction kinetics as they do to binding equilibria (38)–(40); however, this is only so for the bimolecular association process and not for the pseudo-first order dissociation process. Moreover, the situation becomes extra complicated if peptide binding occurs to R-state as well as to S-state Hsp70. For the case of binding to a single state, we have

$$K_D^{APP} = k_{off}/k_{on}^{APP}. \quad (70)$$

Then with reference to Eq. (40), the case for negligible peptide binding to the R state, and to Eq. (41) for corresponding S-state association, we obtain

$$k_{on}^{APP}(ATP) = k_{on}^S(ATP)/\left(K_{eqS} + 1\right). \quad (71)$$

The situation is not so simple in the general case where peptides bind both to R and S states since most experiments will not discriminate.

### Kinetics of substrate peptide release

Having to consider substrate peptide binding to both the R and S states also complicates general considerations on peptide dissociation; however, analyses can be made in certain circumstances.

One particular experiment of interest concerns the measurement of $k_{off}$ for peptide release from DnaK in the presence of ATP. Typically, one incubates DnaK with a labelled peptide, P*, in the absence of nucleotides or in ADP, and then measures the release of labelled peptide after mixing with a solution containing ATP and excess unlabelled peptide at concentration [P]. In general, the chaperone will be in equilibrium between S and R states, from which release will occur according to Eqs. (72) and (73):

$$SP^*(t) \overset{k_{off}^{S}}{\rightarrow} S + P^* [SP^*(t)] = [SP^*]_0 \exp\left(-k_{off}^{S}t\right) \quad (72)$$

$$RP^*(t) \overset{k_{off}^{R}}{\rightarrow} R + P^* [RP^*(t)] = [RP^*]_0 \exp\left(-k_{off}^{R}t\right) \quad (73)$$

Thereby, released labelled peptide accumulates as

$$[P^*(t)] = \left([SP^*(t)] - [SP^*]_0\right) + \left([RP^*(t)] - [RP^*]_0\right)$$

$$[P^*(t)] = [SP^*]_0\left\{1 - \exp\left(-k_{off}^{S}t\right)\right\} + [RP^*]_0\left\{1 - \exp\left(-k_{off}^{R}t\right)\right\}. \quad (74)$$

The equilibrium between the S and R states is governed by Eqs. (1)–(3), whereby the fraction in the stimulating state, Q, is given by Eq. (10) and the remainder, $1 - Q$, is in the restraining state. At $t = 0$, $c_T^*(0) = [SP^*]_0 + [RP^*]_0$. As the dissociation proceeds, the products S and R also accumulate along with P*, but perhaps at very different rates. Nevertheless, since we assume that both S and SP equilibrate identically with respect to both R and RP, $Q([P], t)$ is expected by Eq. (11) to be invariant with time:

$$[P^*(t)] = c_T^*(0)\{Q([P])\left(1 - \exp\left(-k_{off}^{S}t\right)\right) + (1 - Q([P]))\left(1 - \exp\left(-k_{off}^{R}t\right)\right)\}. \quad (75)$$

The sensitivity of $Q([P])$ to peptide concentration is, of course, governed by the overall peptide concentration, which the excess of unlabelled peptide can be assumed to dominate.

A complication with the experiment for peptide dissociation in the presence of ATP is that ATP hydrolysis by Eqs. (13) and (14) will deplete SP and RP components to yield Hsp70$_U$-ADP, from which the labelled peptide P* will then dissociate by Eq. (4) with $k_{off}^{U}$. Taking these two steps into account in Eq. (75) yields Eq. (76):

$$[P^*(t)] = [SP^*]_0\left\{1 - \exp\left(-k't\right)\right\}\left\{1 - \exp\left(-k_{off}^{S}t\right)\right\}$$
$$+ [RP^*]_0\left\{1 - \exp\left(-k^0t\right)\right\}\left\{1 - \exp\left(-k_{off}^{R}t\right)\right\}$$
$$+ \left\{[SP^*]_0 \exp\left(-k't\right) + [RP^*]_0 \exp\left(-k^0t\right)\right\}$$
$$\left\{1 - \exp\left(-k_{off}^{U}t\right)\right\}, \quad (76)$$

and Eq. (64) follows on rearrangement. Notice that while ATP hydrolysis may instantaneously change the relative proportions in S and P states, these will re-equilibrate by (11) to $Q([P])$ with [P] typically dominated by unlabelled peptide in excess of the labelled P*.

$$[P^*(t)] = c_T^*(0)\left\{Q([P])\left[\left(1 - \exp(-k't)\right)\left(1 - \exp\left(-k_{off}^{S}t\right)\right)\right.\right.$$
$$+ \exp(-k't)\left(1 - \exp\left(-k_{off}^{U}t\right)\right)]$$
$$+ (1 - Q([P]))\left[\left(1 - \exp(-k^0t)\right)\left(1 - \exp\left(-k_{off}^{R}t\right)\right)\right.$$
$$\left.\left. + \exp\left(-k^0t\right)\left(1 - \exp\left(-k_{off}^{U}t\right)\right)\right]\right\}. \quad (77)$$

### Special cases

One special case of interest is the fraction in the stimulating state at [P] = 0, which from Eq. (12) is

$$Q(0) = \frac{1}{(K_{eqS} + 1)}. \quad (78)$$

Another special case of interest is the hydrolysis rate at [P] = 0, which from Eq. (17) is

$$k_{cat} = k^0 + \frac{1}{(K_{eqS} + 1)} \times \left(k' - k^0\right). \quad (79)$$

Also from Eq. (17), the hydrolysis rate for the fully stimulated case ($K_{eqS} = 0$) at the limit of $K_{eqS} = \infty$ is $k_{cat} = k'$ and that at the fully restrained state ($K_{eqS} = \infty$) is $k_{cat} = k^0$.

### Correction to obtain free peptide concentration

Although the formulations given by Eqs. (14) and (16) depend on [P], the concentration of free peptide, one is only able to control the total concentration of the peptide under study, which at the outset before ATP hydrolysis is given by Eq. (80):

$$c_P = [P] + [SP]. \quad (80)$$

Then, upon substituting Eq. (9) into Eq. (80) and rearranging the factors, one obtains the quadratic Eq. (81):

$$[P]^2 + \left\{(c_T - c_P) + K_D^{0S}\left(K_{eqS} + 1\right)\right\}[P] - K_D^{0S}\left(K_{eqS} + 1\right)c_P = 0, \quad (81)$$

which can be solved readily for the desired [P] as a function of the experimentally accessible $c_P$. For parameters relevant to WT DnaK, [P] is at 99% of $c_P$ even at $c_P = 5\,\mu M$ and the fraction increases as $c_P$ increases. For the parameters of mutants such as I483D, where $K_{eqS} = 0$, the [P] fraction is reduced to 82% at $c_P = 5\,\mu M$ but reaches 99% by $100\,\mu M$. In other words, for practical situations with Hsp70s, measures of $c_P$ are reasonably close to the free peptide concentration [P].

## Experimental Validation

Our theoretical explication was devised to explain observations that we had made on allosteric phenomena in Hsp70 action, and we use such data to test the formulations. The theory also predicts a previously uncharacterized stimulating state conformation, and we designed constructs that have successfully captured this state.

### ATP hydrolysis controlled by peptide binding

A driving motivation for our theoretical development came in explaining the effect of peptide binding on ATP hydrolysis. The rate of hydrolysis is observed to accelerate as a function of substrate peptide concentration and to be affected by certain mutations. In a

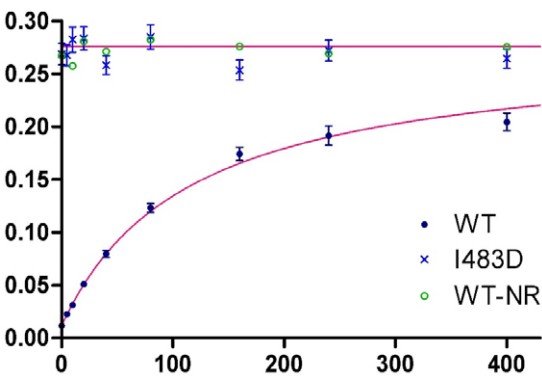

**Fig. 2.** Effect of substrate peptide binding on ATP hydrolysis by Hsp70 DnaK. Observed rates of hydrolysis $k_{cat}$ and standard deviations, as reported elsewhere (Wang *et al.*, 2020), are plotted as a function of the concentration [P] of NR heptapeptide (sequence NRLLLTG) for WT DnaK (●) and for I483D DnaK (x), which is characterized as fully stimulated constitutively. The smooth curve through WT DnaK data is from the least-squares fitting of measured rates by Eq. (21), which gave a' = 0.276 min⁻¹, b' = ATP 1.33 min⁻¹ μM and d' = 115.1 μM. The straight line through points for I483D is at kcat = k' = a', which is the asymptote for the curve fitted to the WT DnaK data. The hydrolysis rates were measured, as reported Wang *et al.* (2020), in assays of single-turnover kinetics (Schrank *et al.*, 2009).

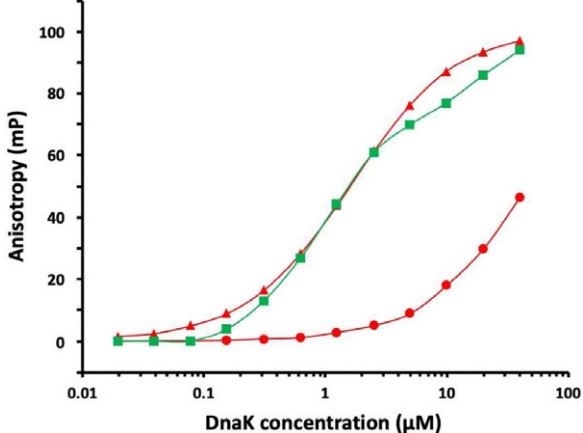

**Fig. 3.** Effects of ATP and ADP on substrate peptide binding by Hsp70 DnaK. The binding of fluorescein-labelled NR peptide (NRLLLTG, 10 nM) was measured by fluorescence anisotropy as a function of DnaK concentration ($c_T$) as described elsewhere (Wang *et al.*, 2020). Measurements are shown for WT DnaK in the presence of ADP (▲) and in the presence of ATP (●) and for I483D DnaK in the presence of ATP (■). Peptide binding to I483D in ADP was indistinguishable from that to WT when in ADP.

separate report (Wang *et al.*, 2020), we describe single-turnover kinetic measurements of ATP hydrolysis by WT Hsp70 DnaK from *E. coli* and by selected mutant variants. Here in Fig. 2, we reproduce the resulting data for WT and I483D DnaK together with fittings based on Eq. (21) with alternative interpretations in terms of allosteric parameters as given by Eqs. (18) and (20). The goodness-of-fit to the WT data is excellent (1.02), giving mathematical parameters a' = 0.276 ± 0.012 min⁻¹, b' = 1.33 ± 0.14 min⁻¹ μM, and d' = 115.1 ± 9.0 min⁻¹ μM. Since our single-turnover $k_{cat}$ measurements are highly accurate, these fitting results provide a stringent test of the theory.

As discussed above, necessarily there is a degeneracy in biochemical parameters of the allosteric model since there are four of these in the formulation of Eq. (20), and five in the formulation of Eqs. (18), (31) and (37) as compared to the three intrinsic mathematical parameters of Eq. (21). Either independent experimental information or *ad hoc* assumptions or approximations are needed to break the degeneracy. We know, for example, that ATP hydrolysis is very slow in absence of peptide; so, if we assume this rate to be negligible, $k^0 = 0$. We also formulated the model based on the premise that peptide affinity in the restraining state is very low; and if we assume it to be negligible, $K_D^{0R} = \infty$. And from our experimental evaluations (Wang *et al.*, 2020), we have deduced that certain mutant variants are in defined states wherein particular biochemical parameters for that mutant should also reflect the WT value. For example, I483D and N170D both appear to be fixed in the stimulating state and these mutations are at sites that would not be expected to affect peptide binding; thus, we might assume $K_D^{0S}$ (WT) = $K_D^{APP}$ (I483D) = $K_D^{APP}$ (N170D). Similarly, although N170D does affect the rate of ATP hydrolysis, I483D would not be expected to do so since this residue is exposed on SBD remote from the catalytic center; thus, by the hydrolysis model of Eq. (15) we might then assume k' (WT) = k' (I483D).

To break the degeneracy here, we extract allosteric parameters from the fitted WT mathematical parameters by first setting $K_D^{0S}$ (WT) = < $K_D^{APP}$ (N170D), $K_D^{APP}$ (I483D) > = < 1.71 ± 0.26, 1.75 ± 0.16 > = 1.73 ± 0.20 μM. With this specification and assuming c = 1, it follows by (22) that $K_{eqS}$ = 65.5 ± 5.2. With c = 1, $K_D^{0R} = \infty$ by

Eq. (23), $k^0 = 0.0075 ± 0.0020$ min⁻¹ by Eq. (27), and k' = a' = 0.276 ± 0.012 min⁻¹ by (28). This fitted value for k' is within experimental error of the rate found constitutively for the S-state mutant I483D, k' = 0.271 ± 0.011 min⁻¹; moreover, the discrepancy is opposite from what could be closed by reducing $K_D^{0R}$ in Eq. (28). For example, if c were increased to 1.1, by (23) $K_D^{0R}$ would decrease only to 1,249 μM, which still implies negligible peptide affinity, while the k' discrepancy would increase from 0.005 to 0.033 min⁻¹ (3.0 σ). Indeed, it is fair to conclude that experiments confirm that model (20) is a valid simplification.

Additional hydrolysis experiments allow us to discriminate among the alternative hydrolysis models. DnaK₆₀₉::NR was constructed with the optimized substrate peptide NRLLLTG fused to DnaK (WT-NR) in a manner disposed for avid binding to the SBD site, thus producing the SP state independent of extrinsic [P] (Wang *et al.*, 2020). The rate of ATP hydrolysis measured for DnaK₆₀₉::NR ($k_{cat}$ = 0.276 min⁻¹) was essentially the same as for the constitutive S-state mutant I483D ($k_{cat}$ = 0.271 min⁻¹), and both are indistinguishable from the WT hydrolytic value, k' = 0.276 ± 0.012 min⁻¹ (Fig. 2). The observation of k'(S) = k'(SP) is as expected from model (15) and inconsistent with model (29) when $k^S$(S) is distinctly different from k', such as being $k^0$; however, these results are compatible with the Q-alternative model (33).

### Peptide binding controlled by ATP binding and hydrolysis

In accord with many other observations, we find that peptide binding to DnaK is much reduced in the presence of ATP as compared to when with ADP (Wang *et al.*, 2020) (Fig. 3). While this is true for WT DnaK, it is not so for mutants that by various biochemical criteria are fixed in the stimulating state, notably N170D and I483D (Wang *et al.*, 2020). With these mutations, DnaK binds peptides with similar affinity whether in the presence of ATP or of ADP. Since the N170D mutation is in NBD and I483D is on an SBD surface remote from the peptide-binding site, we assert that these mutants reflect the intrinsic affinity of the site; that is I483D affinity is high because $K_{eqS}$ = 0 whereby $K_D^{APP}$ = $K_D^{0S}$. The peptide binding affinity for these and other R-state interface mutants are all nearly the same as for WT when in

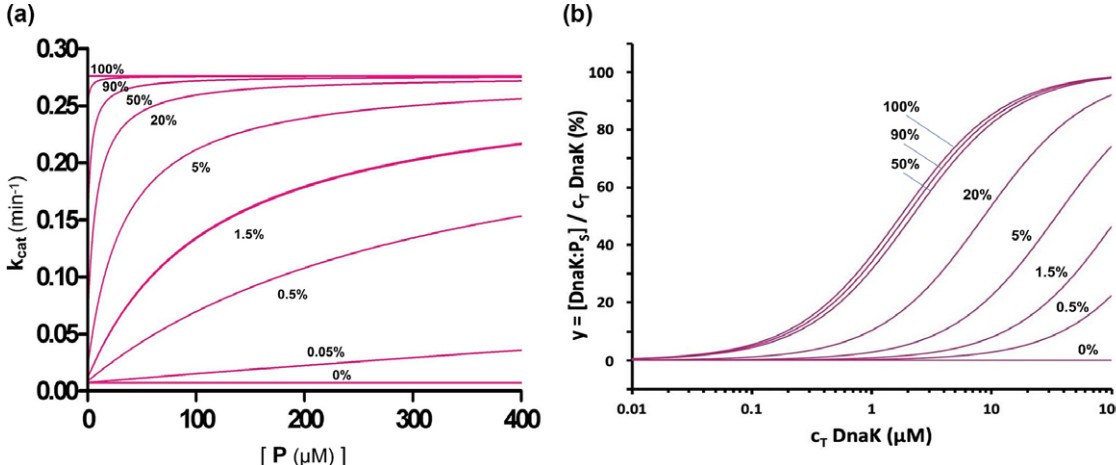

**Fig. 4.** Simulations of effects of varied R vs. S distributions on ATP hydrolysis and substrate peptide binding by Hsp70 DnaK. (*a*) ATP hydrolysis by DnaK as a function of NR substrate peptide concentration. Simulations are by Eq. (20). We assume hydrolytic rate parameters $k' = 0.276$ min$^{-1}$ and $k^0 = 0.0075$ min$^{-1}$ from the fitting in Fig. 1; we assume the intrinsic dissociation constant $K_D^{0S} = 1.73$ μM, the average for fully stimulating mutants I483D and N170D; and we derive the equilibrium constant from Eq. (12), $K_{eqS} = (1 - Q_0)/Q_0$ where $Q_0$ is the specified fraction in the stimulating state at [P] = 0, $Q_S(0)$. $Q_0 = 1.5\%$ for the data in Fig. 1*a*. (*b*) Peptide saturation as a function of DnaK concentration at fixed NR peptide concentration. Simulations are by Eq. (63), assuming [P] = 10 nM, $K_D^{0S} = 1.73$ μM as for *a*, and again obtaining $K_{eqS}$ from the specified $Q_S(0)$ by Eq. (12).

ADP, $K_D$(ADP) $= 1.64 \pm 0.08$ μM and this is nearly the same as the intrinsic affinity with ATP, $K_D^{0S} \equiv < K_D^{APP}$ (N170D, ATP), $K_D^{APP}$ (I483D, ATP) $> = 1.73 \pm 0.20$ μM.

Our experimental results on peptide binding to WT DnaK are compatible with the allosteric theory as formulated in Eq. (39), but not exactly as given by Eq. (40) for the case of $K_D^{0R} = \infty$. By Eq. (40), we calculate $K_D^{APP} = 115.0$ μM using $K_{eqS} = 65.5$ from the fitting to hydrolysis data and $K_D^{0S} = 1.73$ μM as defined by binding to S-state mutants; whereas, we actually measure $K_D^{APP} = 36.7 \pm 5.2$ μM for WT DnaK-ATP (Wang *et al.*, 2020). By Eq. (41), a rearrangement of Eq. (39), we obtain $K_D^{0R} = 53.0$ μM from the measured values for $K_D^{APP}$, $K_{eqS}$ and $K_D^{0S}$; whereas, the fittings to hydrolysis data imply negligible binding to the R-state ($K_D^{0R} > 1$ mM). This dichotomy prompted us to consider the alternative model (42) whereby R is also in equilibrium with a conformation Q that has hydrolysis restrained as for R (33), but which is competent for peptide binding.

The Q-alternative peptide-binding model, Eq. (44), replaces $K_D^{0R}$ with parameters $K_D^{0Q}$ and $K_{eqQ}$ for the newly postulated conformation. Although Eq. (44) has too many variables for independent evaluation, we can obtain the controlling parameters from $K_D^{APP}$ using Eqs. (45) and (46) after assuming values for $K_D^{0S}$ and $K_D^{0Q}$. In this instance, with $K_D^{APP} = 36.7$ μM, we take $K_D^{0S} = 1.73$ μM as before and consider two options for $K_D^{0Q}$. At one plausible extreme, $K_D^{0Q} = K_D^{0S}$ and at another, in light of a Q-like structure with R-like SBD-NBD interfaces and an S-like SBDβ bound to the NR peptide (Wang and Hendrickson, 2020*b*), we use $K_D^{0Q} = 5.66 K_D^{0S}$ as observed comparing a lidless construct to WT DnaK (Buczynski *et al.*, 2001). We obtain $K_{eqS}/K_{eqQ} = 2.14$ and $K_{eqS} = 63.39$ for $K_D^{0Q} = 1.73$ μM and $K_{eqS}/K_{eqQ} = 12.08$ and $K_{eqS} = 53.45$ for $K_D^{0Q} = 9.79$ μM. The distributions among S, Q and R states follow from Eqs. (49)–(55). The fraction in the S-state stays the same as for the R/S model of Eq. (15), which in absence of substrate peptide is $Q_S(0) = 1.5\%$; whereas, the predominating remainder is apportioned differently depending on the Q-state affinity: for $K_D^{0Q} = 1.73$ μM, $Q_Q(0) = 3.2\%$ and $Q_R(0) = 95.3\%$ while for $K_D^{0Q} = 9.79$ μM, $Q_Q(0) = 18.2\%$ and $Q_R(0) = 80.3\%$.

### Simulation of allosteric behaviour

The theory for allosteric regulation of Hsp70 activity permits the possibility to simulate the Hsp70 behaviour under varied conditions. It is of particular interest to consider the impact on chaperone properties of variation in the R–S equilibrium, which is governed by $K_{eqS}$ of Eq. (1). The effects of such variation on ATP hydrolysis are shown in Fig. 4*a* and the effects on peptide binding are shown in Fig. 4*b*. The respective families of curves for varied fractions in the stimulating state as determined by $K_{eqS}$ can be compared with experimental determinations, including those for WT and I483D DnaK given in Figs. 2 and 3, respectively. WT DnaK is dominantly in restraining state R whereas I483D is an extreme mutant fixed in the stimulating state S; other mutants are intermediate. For the I160D mutant, the fitting to peptide-binding data gave $K_D^{APP} = 2.9$ μM and fitting by Eq. (20) to the hydrolysis data gave $K_{eqS} = 4.75$ (17% S-state at [P] = 0); however, this implies $K_D^{APP} = 9.9$ μM by Eq. (40). Exact fitting to the Q-alternative model by Eqs. (45) and (46) assuming $K_D^{0Q} = K_D^{0S}$ is accomplished with a distribution of states $Q_R(0):Q_Q(0):Q_S(0)$ of 40.3%:42.3%:17.4%. By Eq. (47) a physical solution in this case requires that $K_D^{0Q} \leq 1.95 K_D^{0S}$ ($K_D^{0Q} \leq 3.38$ μM), whereby $Q_R(0):Q_Q(0):Q_S(0)$ of 0%:82.6%:17.4%.

### Hsp70 structure

Our initial formulation of a theory on allosteric regulation of Hsp70 molecular chaperones was derived to account for the results of biochemical tests in yeast Ssa1 and *E. coli* DnaK of interface mutations based on the structure of yeast Hsp110 Sse1 as a prototype for Hsp70s (Liu and Hendrickson, 2007; Wang *et al.*, 2020). Structures of hydrolysis-impaired T199A mutants of *E. coli* DnaK in complexes with ATP corroborated the conjecture that ATP-associated Hsp70s would resemble Hsp110 (Kityk *et al.*, 2012; Qi *et al.*, 2013). Moreover, in keeping with Eq. (19) where $K_D^{0R} = \infty$, the peptide-binding sites in these structures are deformed from those in SBD-peptide complexes (Zhu *et al.*, 1996), such as to preclude peptide binding (Kityk *et al.*, 2012; Qi *et al.*, 2013; Wang *et al.*, 2020). In addition, in keeping with the low rate of ATP hydrolysis, $k^0$, as fitted to this predominating restraining R state conformation in the absence of peptide substrates, we find from a series of NBD(ATP) structures that the R state has a portion of NDB, which we call the R-to-S switch segment, in a conformation that blocks hydrolysis (Wang and Hendrickson, 2020*a*) as implied by low $k^0$ in fittings to hydrolysis data.

Perhaps the most important confirmation of the theory is the finding that the alternative S-state conformation, postulated to explain rebinding of peptide substrates and ATP hydrolysis, is found to exist as predicted (Wang *et al.*, 2020). These newly discovered S-state structures have molecular features compatible with biochemical activities. NBD in the S-state has its R-to-S switch segment in a conformation permissive of elevated hydrolysis as implied by higher $k'$ in fittings to hydrolysis data. The NBD-linker construct adopts the S-state conformation when with ATP as expected from the idea that R restrains Hsp70 from hydrolyzing ATP, and when released from SBD interactions it reverts to a potentiated hydrolysis (Wang and Hendrickson, 2020a). SBD in the S-state is receptive to peptide binding and, as seen by the near equivalence of $K_D^{0S}$ to peptide $K_D(ADP)$, the conformation of SBDβ in the S state is nearly identical to that in the uncoupled U state (Wang *et al.*, 2020). SBD in the S-state has the SBDα lid domain flexibly linked in keeping with higher on/off kinetics for WT Hsp70 in ATP as compared to that in ADP or to mutant Hsp70 variants (Wang *et al.*, 2020).

Finally, in keeping with the lack of allosteric coupling in the absence of ATP, an NMR analysis of DnaK(ADP) shows NBD and SBD flexibly linked (Bertelsen *et al.*, 2009), and the contacts between NBD and SBD in X-ray structures of Hsp70s in the presence of ADP (Chang *et al.*, 2008; Adell *et al.*, 2018) or without nucleotide (Jiang *et al.*, 2005) appear to be unnatural and nonproductive interactions, for example. SBD-linker lattice contacts or disordered domains.

## Discussion

Biochemical properties of structure-inspired mutations of interfaces between domains in Hsp70 DnaK prompted the hypothesis that the state of Hsp70 first recognized by analogy to our structure of Sse1-ATP (Liu and Hendrickson, 2007) and also seen in the structures of DnaK$_R$-ATP (Kityk *et al.*, 2012; Qi *et al.*, 2013) is restrained against its hydrolysis of ATP. The theory that we devised to explain such biochemical behaviour provides a sound basis for understanding allostery in Hsp70s. This theory is reminiscent of the famous MWC allosteric equilibrium model developed to explain oxygen binding by hemoglobin (Monod *et al.*, 1965), and it builds from decades of studies of allostery (Monod *et al.*, 1965; Koshland *et al.*, 1966; Cui and Karplus, 2008; Motlagh *et al.*, 2014; Cuendet *et al.*, 2016; Thirumalai *et al.*, 2019); however, whereas MWC controls one binding activity in an oligomer through alternative quaternary states, here two activities are controlled reciprocally through alternative conformations adopted between domains of a single chain. We postulate an allosteric equilibrium between two ATP states: a restraining state with negligible affinity for polypeptide substrates and very limited ATP hydrolysis, and a stimulating state that hydrolyzes ATP readily and binds substrate peptides with rapid exchange kinetics. In the absence of peptide substrates, the restraining state dominates in the equilibrium, and the apparent peptide affinity is much reduced from its intrinsic value. When substrates are present, the equilibrium is drawn to the stimulating state, enhancing ATP hydrolysis and capturing valid substrates in the ADP state. The DnaK$_R$-ATP structures epitomize the restraining state (Kityk *et al.*, 2012; Qi *et al.*, 2013) and our new structures of DnaK$_S$-ATP model depict the stimulating state (Wang *et al.*, 2020).

To be tractable, the theoretical model for peptide-stimulated hydrolysis of ATP needed to be simple. While our model seems to capture the essence of allosteric control in DnaK quite well, the reality may be more complex. For example, whereas we assume that rates of hydrolysis are the same by SP and S, stimulating-state Hsp70 with and without bound peptide, these rates likely differ somewhat. Moreover, the biochemical complexity of the system forced us to consider models, as for ATP hydrolysis experiments, that have more parameters than the resulting data can define unambiguously. In order to break the consequent degeneracy of parameters for such experiments, we have used observations from other experiments to define certain parameters, for example, intrinsic peptide affinity; however, underlying assumptions of equivalence may not hold perfectly.

Our studies have employed cellular and biochemical analyses of mutant variants to test functional hypotheses; however, mutated proteins are imperfect reporters of native function since intended perturbations of activity may extend to unanticipated effects. For example, whereas certain other mutations such as I483D seem to preclude the restraining state and to give a valid picture of stimulating state properties, there may be unanticipated consequences as well. Still, despite shortcomings, such mutation analyses have generated new insights and provided critical tests of Hsp70 function.

The treatment here for allosteric regulation is novel, to the best of my knowledge, both for Hsp70s in particular and for allosteric systems more generally. The result that most closely approaches our Hsp70-ATP equilibrium model (Eqs. 1–3) came from an NMR study (Zhuravleva *et al.*, 2012) showing that DnaK-ATP can have two alternative conformations: an "ATP-bound, domain-docked state" modeled on Sse1-ATP (Liu and Hendrickson, 2007), now known to be very similar to restraining-state DnaK$_R$-ATP (Kityk *et al.*, 2012; Qi *et al.*, 2013, Wang *et al.*, 2020), and an "allosterically active state" bound to both ATP and substrate peptide. With clarifications from additional experiments (Lai *et al.*, 2017), the latter likely relates to our stimulating-state DnaK$_S$-ATP; however, this "partially docked state" is insufficiently specified for direct comparison with our crystal structure results (Wang *et al.*, 2020). For allosteric systems more generally, the simplified case of Eq. (40) on binding, where $K_D^{0R} = K_D^{0Q} = \infty$, has been described before in the context of studies on an adenylate kinase (Schrank *et al.*, 2009); however, we are not aware of previous formulations comparable to Eqs. (17)–(20), (36) and (37) on hydrolysis or to Eqs. (39) and (44) on binding for any system. An empirical fitting to DnaK hydrolysis data does have a form equivalent to Eq. (20) (Slepenkov and Witt, 2002).

## Methods

### Fitting to conformational equilibrium theory

We developed a least-squares program for the fitting of parameters to our theoretical model of peptide-stimulated ATP hydrolysis by Hsp70 proteins. The observed $k_{cat}$ values are weighted by their inverse variances, $w = 1/\sigma^2$, where $\sigma = (\sigma_{fit}^2 + \sigma_{sys}^2)^{1/2}$ where $\sigma_{fit}$ is the random error deduced from fitting to the kinetic data and $\sigma_{sys}$ is a systematic error increment to account for added variations in repeated constant measurements, as for N170D, V389D and I483D at varying peptide concentrations. We find $\sigma_{sys} = 0.0324 \times k_{cat}$.

ATP hydrolysis and peptide binding measurements were made and analyzed as described in a companion paper (Wang *et al.*, 2020).

**Open Peer Review.** To view the open peer review materials for this article, please visit http://doi.org/10.1017/qrd.2020.10.

**Acknowledgements.** I thank Wei Wang, Qun Liu, and Qinglian Liu for discussions about Hsp70 conformations, which were critical to the formulation and testing of the theory, and I thank Qinglian Liu for use of her data on ATP hydrolysis and substrate peptide binding by Hsp70 DnaK in experimental

validations of the theory. This work was supported in part by NIH grant GM107462.

**Conflict of Interest.** The author declares no competing financial interests.

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
