## [Reviewer Report]

*Comments to Author*: I have no substantial criticism of the theory being advanced. However, the paper would be vastly improved if the authors first submitted the structure to the PDB. The absence of structural information is very strange indeed.

---

## [Reviewer Report]

*Comments to Author*: This is an excellent paper which I recommend for publication in QRB Discovery. There is clearly need for a theoretical framework to interpret regulation in Hsp70 Molecular Chaperones, here addressing heat-shock proteins a very pertinent class of problems. The paper reads well and I may only wish that the rather complex system of equilibria were commented on in a little more pedagogical way as to dominant effects and impact.

---

## [Reviewer Report]

*Comments to Author*: My overall opinion of Wayne’s manuscript is positive, as I often react to the excellent work that is produced by the Hendrickson lab. In this case, the need for a theoretical framework to interpret Allosteric Regulation in Hsp70 Molecular Chaperones is very much justified to rationalize a large body of experimental observations. In this manuscript, Wayne accomplishes this goal for Heat-shock proteins of 70 kDa (Hsp70s). Specifically, the model described here for Hsp70 allostery evokes equilibria among Hsp70 conformational states. At the heart of the formalism is the expectation that upon binding of ATP, Hsp70 equilibrates, in the author’s words ".. between a restraining state (R) that restricts ATP hydrolysis and binds peptides poorly, if at all, and a stimulating state (S) that hydrolyzes ATP relatively rapidly and has high intrinsic substrate affinity but rapid binding kinetics; after the hydrolysis to ADP, NBD and SBD disengage into an uncoupled state (U) that binds peptide substrates tightly but now with slow kinetics of exchange." This creative model is plausible, if not singularly unique, and deserves to be vetted in the literature. I recommend publication.

---

## [Reviewer Report]

*Comments to Author*: General Comments

It is my opinion that the model should be shown as a detailed mechanism which may greatly help the reader, including myself. Such a mechanism, which was written based on the information found in the paper, was sent to the Editor as it can not be uploaded here. I have also named the species differently to simplify the nomenclature and clearly distinguish all states. The equilibrium constants should be indicated for each equilibrium. Needless to say, I do not know if this model is correct. Nevertheless the steady-state solution yields eq. 18 but not 20. The procedure involves: 1) writing the mechanism containing n species related by reactions; 2) writing the mass conservation of the chaperone; 3) writng n-1 equations containing the equilibrium constants expressions; 4) solving the n simultaneous equation. One thus obtains the steady-state concentrations of all species. The expression of the initial rate (a function of T, D, P, total chaperone, equilibrium constants and k0,k’) is therefore obtained. In my nomenclature this is

v0 =k0*[RT]ss+k0*[RTP]ss+k’*[ST]ss+k’*[STP]ss (where ss stands for steady-state).

Other comments.

1) this is clearly a very complex system with many rate constants and corresponding equilibirum constants. It is my view that it is hardly testable unless simplyfing experimental conditions are found. Is it possible, for example, to "freeze" the chaperone in the R or S states? Other?

2) The proposed model was solved under the so-called quasi-steady-state assumption (i.e. the time derivatives of all species concentrations set to zero) with the further restriction that all mechanism reactions are at equilibrium with respect to the hydrolysis reactions described by k0 and k’. Is there any evidence that this is the case?

3) the Author refers to the apo protein as a species devoid of ATP, ADP, substrate peptide and phosphate (not mentioned in the paper). The term "apo", however, refers to the protein part of an enzyme lacking its characteristic prosthetic group.

4) it is not clear if the constant kcat refers to the initial rate or the turnover number.

---

## [Reviewer Report]

*Comments to Author*: Reviewer #1: My overall opinion of Wayne’s manuscript is positive, as I often react to the excellent work that is produced by the Hendrickson lab. In this case, the need for a theoretical framework to interpret Allosteric Regulation in Hsp70 Molecular Chaperones is very much justified to rationalize a large body of experimental observations. In this manuscript, Wayne accomplishes this goal for Heat-shock proteins of 70 kDa (Hsp70s). Specifically, the model described here for Hsp70 allostery evokes equilibria among Hsp70 conformational states. At the heart of the formalism is the expectation that upon binding of ATP, Hsp70 equilibrates, in the author’s words ".. between a restraining state (R) that restricts ATP hydrolysis and binds peptides poorly, if at all, and a stimulating state (S) that hydrolyzes ATP relatively rapidly and has high intrinsic substrate affinity but rapid binding kinetics; after the hydrolysis to ADP, NBD and SBD disengage into an uncoupled state (U) that binds peptide substrates tightly but now with slow kinetics of exchange." This creative model is plausible, if not singularly unique, and deserves to be vetted in the literature. I recommend publication.

Reviewer #2: General Comments

It is my opinion that the model should be shown as a detailed mechanism which may greatly help the reader, including myself. Such a mechanism, which was written based on the information found in the paper, was sent to the Editor as it can not be uploaded here. I have also named the species differently to simplify the nomenclature and clearly distinguish all states. The equilibrium constants should be indicated for each equilibrium. Needless to say, I do not know if this model is correct. Nevertheless the steady-state solution yields eq. 18 but not 20. The procedure involves: 1) writing the mechanism containing n species related by reactions; 2) writing the mass conservation of the chaperone; 3) writng n-1 equations containing the equilibrium constants expressions; 4) solving the n simultaneous equation. One thus obtains the steady-state concentrations of all species. The expression of the initial rate (a function of T, D, P, total chaperone, equilibrium constants and k0,k’) is therefore obtained. In my nomenclature this is

v0 =k0*[RT]ss+k0*[RTP]ss+k’*[ST]ss+k’*[STP]ss (where ss stands for steady-state).

Other comments.

1) this is clearly a very complex system with many rate constants and corresponding equilibirum constants. It is my view that it is hardly testable unless simplyfing experimental conditions are found. Is it possible, for example, to "freeze" the chaperone in the R or S states? Other?

2) The proposed model was solved under the so-called quasi-steady-state assumption (i.e. the time derivatives of all species concentrations set to zero) with the further restriction that all mechanism reactions are at equilibrium with respect to the hydrolysis reactions described by k0 and k’. Is there any evidence that this is the case?

3) the Author refers to the apo protein as a species devoid of ATP, ADP, substrate peptide and phosphate (not mentioned in the paper). The term "apo", however, refers to the protein part of an enzyme lacking its characteristic prosthetic group.

4) it is not clear if the constant kcat refers to the initial rate or the turnover number.

Reviewer #3: This is an excellent paper which I recommend for publication in QRB Discovery. There is clearly need for a theoretical framework to interpret regulation in Hsp70 Molecular Chaperones, here addressing heat-shock proteins a very pertinent class of problems. The paper reads well and I may only wish that the rather complex system of equilibria were commented on in a little more pedagogical way as to dominant effects and impact.

Reviewer #4: I have no substantial criticism of the theory being advanced. However, the paper would be vastly improved if the authors first submitted the structure to the PDB. The absence of structural information is very strange indeed.